# Coupled Precipitation of Dual-Nanoprecipitates to Optimize Microstructural and Mechanical Properties of Cast Al–Cu–Mg–Mn Alloys via Modulating the Mn Contents

**DOI:** 10.3390/nano13233038

**Published:** 2023-11-28

**Authors:** Han Zhang, Qitang Hao, Xinlei Li, Wentao Yu, Yanqing Xue

**Affiliations:** 1State Key Laboratory of Solidification Processing, Northwestern Polytechnical University, Xi’an 710072, China; laraine0222@163.com (H.Z.); lxl@nwpu.edu.cn (X.L.); yqxue666@163.com (Y.X.); 2Shaanxi Key Laboratory of Surface Engineering and Remanufacturing, Xi’an University, Xi’an 710065, China; yuwentao417@163.com

**Keywords:** Al–Cu–Mg–Mn alloys, nanoprecipitates, microstructure, mechanical properties, strengthening mechanisms

## Abstract

The effect of Mn content on the microstructure evolution and mechanical properties of Al–Cu–Mg–*x* Mn alloys at ambient temperature was investigated. The findings show that in the Mn-containing alloys at the as-cast state, the blocky primary T(Al_20_Cu_2_Mn_3_) phase coexisting with the Al_2_Cu phase appeared. With the increase in Mn content, the majority of the Al_2_Cu phase dissolved, nd a minor amount of the T phase remained at the grain boundary after solution treatment. The rod-like T_Mn_ (Al_20_Cu_2_Mn_3_) nanoprecipitate was simultaneously distributed at grain boundaries and the interiors, while a high density of needle-like θ″ (Al_3_Cu) nanoprecipitate was also observed in the T6 state. Further increases in Mn content promoted the dispersion of the T_Mn_ phase and inhibited the growth and transformation of the θ″ phase. Tensile test results show that 0.7 wt.% Mn alloy had excellent mechanical properties at ambient temperature with ultimate tensile strength, yield strength, and fracture elongation of 498.7 MPa, 346.2 MPa, and 19.2%, respectively. The subsequent calculation of strengthening mechanisms elucidates that precipitation strengthening is the main reason for the increase in yield strength of Mn-containing alloys.

## 1. Introduction

Al–Cu–Mg alloys, known for their high strength, excellent fatigue resistance, and lightweight nature, exhibit potential engineering applications at ambient temperatures in the aircraft and automobile industries [1,2,3,4]. However, the development of modern industry has resulted in new requirements for aluminum alloy engineering components, like trade-offs of high strength and ductility. The dominant strengthening approach for Al–Cu–Mg alloys is precipitation strengthening, in which the precipitation sequence is closely dependent on the Cu/Mg ratio [5]. At high Cu/Mg ratios, the θ (Al_2_Cu) nanoprecipitate precipitated via the precipitation sequence: supersaturated solid solution (SSSS) → Guinier-Preston (GP) zones → θ″ (Al_3_Cu) → θ′ (Al_2_Cu) → θ (Al_2_Cu) [6,7]. Further increases in the strength of cast Al–Cu–Mg alloys of high Cu/Mg ratios are hardly achieved by adjusting precipitation strengthening via individual modulation of Cu/Mg ratios [8].

Microalloying has been considered an effective strategy for improving the comprehensive mechanical properties of Al–Cu–Mg alloys, and the relevant alloying elements can be classified into two types: the fast-diffusing ones and the slow-diffusing ones. The former can be exemplified by Si, Zn, Ag, etc. For example, a small amount of Si facilitates the formation of the Q (Al_4_Cu_2_Mg_8_Si_7_) phase and refines the S (Al_2_CuMg) nanoprecipitate (orthorhombic crystal structure with lattice parameters a = 0.400 nm, b = 0.923 nm, c = 0.714 nm) in the Al–Cu–Mg alloys [9]. Adding Zn to the Al–Cu–Mg alloys, the Zn/Mg ratio has proven to be an important factor in regulating the precipitation. Generally, the Al_2_Mg_3_Zn_3_ phase is the major strengthening phase when the Zn/Mg ratio ≤ 1, while the η (MgZn_2_) phase is the dominant strengthening phase when the Zn/Mg ratio ≥ 2; these two phases can effectively enhance the strength [10]. Bai et al. [11] found that microalloying with Ag, due to the strong binding energy between Ag and Mg, induced the precipitation of fine-plate Ω (Al_2_Cu) nanoprecipitate (orthorhombic crystal structure with lattice parameters a = 0.496 nm, b = 0.859 nm, c = 0.848 nm), which enhance the thermal stability of the Al–Cu–Mg alloys. The latter counterparts represented by Sc, Zr, and Y with sluggish diffusivity have also been proven to considerably improve ambient and elevated temperature strengths by forming coarsening-resistant particles (such as Al_3_X nanoprecipitates) in cooperation with the conventional nanoprecipitates (such as θ′ nanoprecipitates—a bct structure with lattice parameters a = 0.404 nm and c = 0.580 nm [12]). For instance, the trace addition of Sc to the Al–Cu–based alloys remarkably promote the homogeneous formation of the θ′ phase and limits the growth of the θ′ phase, which is achieved through the Sc element segregation at the θ′/α–Al interfaces [13]. Etl et al. [14] added Sc and Zr to 2219 alloy, which reached a high value of 536 MPa at ambient temperature by assembling Al_3_(Sc, Zr) and θ′ precipitates. Mei et al. [15] demonstrated that after adding Y in the Al–Cu–Mg–Ag alloy, the tensile strength at 300 °C improved significantly due to the segregation of the Al_8_Cu_4_Y phase at the grain boundary. However, the tensile strength decreased remarkably at ambient temperature owing to the inhibition of the precipitation of the Ω phase. Recently, the cooperative addition of Sc and Zr has been confirmed to significantly improve the coarsening resistance of the θ′ phase through interfacial segregation [16,17]. However, this effect may be limited due to the low diffusivity of Sc and Zr at conventional artificial aging temperatures. A recent study has reported that Mn, Fe, and Co have the prominent driving force for segregating at the θ′/α–Al interfaces in accordance with the density functional theory (DFT) and thus are potential candidates for stabilizing the θ′ phase [18,19]. Fu et al. [8] have demonstrated that Mn microalloying stimulates the formation of rod-like T_Mn_ nanoprecipitates in Al–Cu–Mg alloys during solid solution formation, which significantly improves the mechanical properties. 

Previous studies have been focused on revealing the crystal structure of the T_Mn_ phase (orthorhombic structure with lattice parameters a = 2.42 nm, *b* = 1.25 nm, and *c* = 0.775 nm) [20,21] and its interface relationship with the matrix, but few investigations have been emphasized the impact of the T_Mn_ phase on mechanical properties [22,23,24]. The two conspicuous benefits of Mn addition to the Al–Cu–Mg–*x* Mn alloys are summarized as follows: (i) a multitude of T_Mn_ particles formed during homogenization can impede the movement of dislocations to improve strength [25,26], (ii) the distinct formation temperature between the T_Mn_ and conventional precipitates overcomes the adversity of the synchronous precipitation of dual-strengthening precipitates [17]. However, the effect of Mn microalloying on the microstructure and mechanical properties of the Al–Cu–Mg alloys remains to be explained in two aspects: (i) Excessive Mn can lead to the formation of coarse Mn-rich intermetallic compounds during solidification, which affects ductility [27,28]. (ii) The formation of harmful intermetallic compounds and the precipitation of abundant T_Mn_ particles consume Cu solute, which reduces the precipitation driving force of aging precipitates, thereby affecting strength. Therefore, determining the appropriate Mn content as a solution to balance favorable and unfavorable factors is necessary.

Based on the aforementioned discussion, we systematically investigated the effect of Mn content on the microstructure and ambient mechanical properties of the quaternary Al–Cu–Mg–Mn alloys to determine the optimal composition. This work aims to modulate the precipitation of nanoscale T_Mn_ particles and θ″ precipitates by adding appropriate Mn to improve the ambient temperature strength of the Al–Cu–Mg–*x* Mn alloys. The strengthening mechanism of alloys with different Mn contents was elucidated, which is beneficial to the improvement in Al–Cu–Mg–*x* Mn alloys to meet the high strength requirements of industrial applications.

## 2. Materials and Experimental Details

### 2.1. Alloy Preparation and Heat Treatment

Al–Cu–Mg–*x* Mn alloys (*x* = 0 wt.%, 0.5 wt.%, 0.7 wt.%, 0.9 wt.%, and 1.1 wt.%) were prepared by melting high-purity Al ingots, Al–50Cu, Al–10Mn, and Al–10Mg (wt.%) alloys in a resistance furnace of air atmosphere with a graphite crucible, which was coated with zinc oxide. The chemical compositions of the experimental alloys, determined by inductively coupled plasma (ICP 7600, Thermo Fisher Scientific, MA, USA), are shown in Table 1. Initially, the raw materials were heated to 750 °C to form a melt and kept for 40 min until completely melted. The melt was then cooled to 700 °C, and Al–10Mg ingot was added and stirred for 10 min. After that, the temperature of the melt was raised to 730 °C, and a refining agent (0.5% C_2_Cl_6_) was used to purify the melt for 10 min. Following cooling down to 705 °C, the melt was poured into a mold preheated at 210 °C. The ingots of the five alloys were solution treated at 530 °C for 12 h to dissolve the primary phases formed at non-equilibrium solidification, followed by water quenching to room temperature, and then aged at 175 °C for 4 h. 

### 2.2. Mechanical Testing

The bar-shaped samples with a gauge length of 25 mm and a diameter of 5 mm for the tensile test were extracted from the central section of castings in accordance with ASTM B557-84 [29]. Subsequently, uniaxial tensile tests were performed on an electronic universal material testing machine-INSTRON 3382 (Instron Corprration, Canton, MA, USA) at a rate of 5 × 10^−4^ s^−1^. In order to precisely measure the elastic deformation, we have used a 25 mm extensometer to monitor the strain during tensile deformation. The authenticity and reproducibility of the results have been ensured by carrying out a minimum of three tests on each sample of the composition.

### 2.3. Microstructure Characterization

The samples for microstructure observation were prepared using mechanical grinding. Scanning electron microscopy (SEM, ZEISS Gemini 500, ZEISS, Oberkochen, Germany) equipped with energy-dispersive spectroscopy (EDS) by Oxford Instruments (Concord, MA, USA) and high-resolution transmission electron microscopy (HRTEM, FEI Talos F200, FEI company, Hillsboro, OSU, USA) operating at 200 KV with scanning transmission electron microscopy (STEM) were used to characterize the microstructure of the Al–Cu–Mg–*x* Mn alloys. The specimens for SEM were prepared by grinding sequentially with 400#, 1000#, 1500#, 2000#, and #3000 SiC sandpapers and gradually polished to mirror surface. The specimens for TEM were first ground by mechanical polishing to a thickness of 60 μm, and then 3 mm diameter discs were taken from the specimens and subjected to ion milling utilizing a precision ion polishing system (Gatan 691, Pleasanton, CA, USA) with an ion gun beam energy of 4.2 KeV. Electron back-scattered diffraction (EBSD, Oxford instruments, Bicester Village, UK) was used to characterize the grain size of the investigated alloys. For EBSD analysis, electropolishing was performed at −20 °C using an electrolyte comprising 90% C_2_H_5_OH and 10% HClO_4_ by volume. Areas of approximately 800 × 800 µm^2^ were scanned at a step size of 0.2 μm on a FEG-SEM scanner equipped with an EBSD detector. Analyze data with the assistance of AZtec and Channel 5.0 software.

The quantitative analysis of precipitates was obtained by measuring the mean width and length of at least 500 precipitates with the aid of the Image Pro Plus software. The volume fraction *f* of the precipitates was calculated by the following formula [30]: $f=\frac{\pi N_{v}D_{p}^{2}t_{p}}{4}$, where $N_{v}$ is the number density of precipitates, and $t_{p}$ is the mean thickness (mean width). $D_{p}$ is the mean true diameter associated with the mean diameter $D_{m}$ (mean length) by the following relationship: $D_{p}=\frac{2\left(D_{m}-t\right)+2\sqrt{{\left(D_{m}-t\right)}^{2}+\pi D_{m}t}}{\pi }$ [30,31]. The number density $N_{v}$ of the precipitates was estimated by $N_{v}=N_{p}\cdot \frac{\left(1+\frac{D_{p}+t}{2\sqrt{A_{s}}}\right)}{A_{s}\left(D_{p}+t\right)}$, where $N_{p}$ is the number of counted precipitates, *t* is the foil thickness, which was measured by convergent beam electron diffraction technique, and $A_{s}$ is the area of TEM micrographs.

## 3. Results

### 3.1. As-Cast Microstructure

Figure 1 shows the backscattered electron image (BSE) of the as-cast Al–Cu–Mg–*x* Mn alloys with various Mn contents. The as-cast microstructure of the alloys consists of the α–Al dendritic and the grain boundaries, which are mainly dominated by a white non-equilibrium phase, a grey blocky phase, and a fibrous phase. By analyzing the ternary phase diagrams of Al–Cu–Mn and Al–Cu–Mg alloys [32,33,34], the three possible eutectic reactions are L → α(Al) + θ(Al_2_Cu) (548 °C), L → α(Al) + θ(Al_2_Cu) + T(Al_20_Cu_2_Mn_3_) (547.5 °C), and L → α(Al) + θ(Al_2_Cu) + S(Al_2_CuMg) (508 °C), respectively, according to the theoretical solidification temperatures from high to low. Combined with the EDS results shown in Figure 1, it can be concluded that the grey blocky phase is the T phase enriched with Al, Cu, and Mn elements, and the white non-equilibrium phase is the primary θ phase enriched with Al and Cu elements. With the increase in Mn content, there are noticeable differences in the elemental distribution of Cu and Mn. Cu, as a major element, combined with Mn to form the eutectic phase, which is gradually rejected to grain boundaries or intergranular during solidification. Taking the 0.7 wt.% Mn alloy as an example, the fibrous phase beyond the T and θ phases was further analyzed using TEM and EDS. The high-angle annular dark-field (HAADF) image and corresponding EDS results are shown in Figure 2a–e. Element distribution mappings show that the fibrous phase is rich in Al, Cu, and Mg. Therefore, it can be confirmed that the fibrous phase is the S phase (marked by red arrows in Figure 2a) formed during solidification.

With the addition of Mn, the Al_2_Cu phase at the grain boundaries becomes finer, the number of T phases witnessed a remarkable increase, and the grains are refined to some extent. Considering the segregation of dendrites during the solidification process in the Al–Cu–Mg–Mn alloys, eutectic reactions occur with the growth of α–Al and an increase in the concentration of solute elements in the remaining liquid phase, including Cu, Mn, and Mg. During solidification, the T phase is formed within the Al–Cu–Mg–Mn alloy and gradually pushed towards the inter-dendritic or grain boundaries. When Mn content is at a moderate level, the Mn-riched eutectic phase is gathered in the liquid phase at the front of the liquid-solid interface, leading to an increase in constitutional subcooling. As a result, the driving force for solidification is increased, which facilitates grain refinement [8,35].

### 3.2. T6-State Microstructure

The BSE images of the T6-state Al–Cu–Mg–*x* Mn alloys are presented in Figure 3. The T6-state microstructure of the studied alloys is composed of an α–Al matrix, the eutectic T phase, and the granular T_Mn_ precipitate, which are distributed near the grain boundary (Figure 3). The eutectic T phase marked by white arrows is formed at the grain boundaries during solidification, whereas the T_Mn_ phase marked by yellow arrows is precipitated as a fine dispersion during the solution treatment. With the increase in Mn content, the distribution of the T_Mn_ phase becomes wider and more uniform, which causes the T_Mn_ phase to precipitate increasingly and enrich from the grain boundary to the grain interior after solution treatment. Notably, the undissolved eutectic T phase after solution treatment increases in 0.9 wt.% and 1.1 wt.% Mn alloys.

Figure 4a,b show the morphology of nanoscale precipitates in 0.7 wt.% and 1.1 wt.% alloys in the T6 state. As shown in Figure 4a,b, rod-shaped precipitates with a width of approximately 50–130 nm and a length of approximately 200–1000 nm are diffusely distributed inside the grains. The cross-section of the rod-shaped precipitates with the corresponding selected-area electron diffraction (SAED) pattern shown in Figure 4c exhibits the morphological characteristics and structure of multiple twins, which are consistent with the characteristics of T_Mn_ precipitates. The EDS results shown in Figure 4d–h indicate that the rod-shaped precipitates are rich in Al, Cu, and Mn. Furthermore, the result of point compositional analysis (Al, 79.67 at.%; Cu, 7.73 at.%; and Mn, 12.6 at.%) demonstrates that the composition of the rod-shaped precipitates is consistent with the chemical formula of the T_Mn_ phase. The quantitative results of precipitates for the Al–Cu–Mg–*x* Mn alloys subjected to different Mn contents are tabulated in Table 2. Statistical data shows that the mean length and thickness of the T_Mn_ phase are approximately 598.5 nm and 112.0 nm for 0.7 wt.% Mn alloy and 576.4 nm and 124.7 nm for 1.1 wt.% Mn alloy, respectively. In addition, another remarkable feature of 1.1 wt.% Mn alloy is the higher volume fraction of the T_Mn_ phase compared with 0.7 wt.% Mn alloy (6.01% vs. 4.23%, Figure 4a,b).

Figure 5 presents representative BF images of the Al–Cu–Mg–*x* Mn alloys subjected to different Mn contents. The high density of θ″ phase, which can be demonstrated by bright discontinuous streaks passing through the {200}_Al_ diffraction spots shown in SAED patterns (see inserts of Figure 5a–c), is uniformly precipitated in the matrix. The θ″ phase in the Mn-free alloy has a mean length of 34.3 nm and a width of 2.7 nm. No significant difference in the size of the θ″ phase is observed between the 0.7 wt.% Mn and 1.1 wt.% Mn alloys. However, the volume fraction of θ″ precipitates decreased from 1.88% for the Mn-free alloy to 1.04% for the 0.7 wt.% Mn alloy and 0.65% for the 1.1 wt.% Mn alloy. This phenomenon reveals that the consumption of the Cu solute caused by the formation of T_Mn_ particles decreases the precipitation driving force of the θ″ phase, thereby inhibiting the nucleation of the θ″ phase, which is reflected in the reduced number density of θ″ precipitates. Overall, the addition of Mn achieves synergistic precipitation of T_Mn_ and θ″ phases, but there is a mutual influence relationship between the volume fractions of the two phases.

### 3.3. Mechanical Properties

Figure 6a shows the tensile stress-strain curves, and the mechanical properties of the T6-state Al–Cu–Mg–*x* Mn alloys at ambient temperature are presented in Figure 6b. As the Mn content increases, the ultimate tensile strength (UTS) and yield strength (YS) of the Al–Cu–Mg–*x* Mn alloys show a trend of increasing first and then decreasing. The maximum UTS and YS are achieved by 0.7 wt.% Mn alloy, reaching 498.7 MPa and 346.2 MPa, respectively, which are 65.2 MPa and 39.1 MPa higher than those of Mn-free alloy. As Mn addition increases to 1.1 wt.%, the UTS and YS decrease to 489.2 MPa and 334.7 MPa, respectively, compared with those of 0.7 wt.% alloy. For fracture elongation, 0.7 wt.% Mn alloy reaches a maximum fracture elongation of 19.2%, exhibiting evident strength and fracture elongation balance, whereas 0 wt.% Mn alloy has the lowest fracture elongation of 10.9%. When the Mn content exceeds 0.7 wt.%, the strength and fracture elongation decrease, which could be attributed to the appearance of the coarse eutectic T phase in the microstructure. In general, the addition of Mn can increase the mechanical properties of the Al–Cu–Mg–*x* Mn alloys at ambient temperature, in particular, the ultimate tensile strength and fracture elongation.

### 3.4. Fracture Surface Characterizations

The fracture morphologies of the experimental alloys with different Mn contents after tensile tests are shown in Figure 7. The fracture morphology of the Mn-free alloy consists of ridges and large dimples with aggregation of particles at the bottom, which are confirmed as Al_2_Cu phases by the EDS results (see Figure 7f). Based on the fracture morphologies shown in Figure 7b–e, Mn-added alloys display a host of uniform and fine dimples, accompanied by dispersed particles at the bottom of the dimples, indicating that the ductility of Mn-containing alloys is better than that of Mn-free alloys. However, as the Mn content increases to 0.9 wt.% and 1.1 wt.%, the fracture morphology appears with the characteristics of intergranular fracture, and the dimples become larger and shallower with the fragmented particles at the bottom, which proved to be eutectic T phases according to the EDS result shown in Figure 7f. The coarse eutectic T phase adversely affects the mechanical properties in two ways: On the one hand, it reduces the fracture elongation of the alloys with high Mn content because it serves as a source of cracks and cuts the continuity of the matrix. On the other hand, the Cu consumption for the formation of the eutectic T phase decreases the solid solubility of Cu, thereby reducing the number density of precipitated strengthening phases θ″ resulting in the strength reduction in the alloy with high Mn content. This finding corresponds to the aforementioned mechanical properties and microstructural analysis results.

## 4. Discussion

### 4.1. Effects of Different Mn Contents on the Precipitation Behavior

Generally, a great number of vacancies remained due to the solid solution and quenching procedures, which can combine with the solute atoms to create solute-vacancy clusters. Subsequently, these clusters can serve as heterogeneous nuclear substrates to accelerate the formation of precipitates. However, Mn addition facilitates the precipitation of a large number of T_Mn_ particles at the solid solution and quenching process, which reduces the volume fraction of θ″ precipitates after aging treatment. This phenomenon is due to the following reasons: the primary Al_2_Cu phase at the grain boundary is dissolved, and the Cu atoms are initially enriched near the grain boundary during the solidification; further, as the Cu atoms diffuse to the interior of grains, the Mn atoms inside the grain combine with the Cu atoms to form the finely diffused T_Mn_ phase. The number of T_Mn_ phases inside the grain is reduced or even absent due to the continuous consumption of the supersaturated Cu solute, causing a decrease in the concentration gradient from the grain boundary to the interior. Furthermore, the consumption of the Cu solute caused by the formation of T_Mn_ particles decreases the precipitation driving force of the θ″ phase, thereby inhibiting the nucleation and growth of the θ″ phase, as demonstrated by the observed reduction in the number density of θ″ precipitates.

### 4.2. Effects of the Dislocation-Precipitate Interaction on Mechanical Properties

Figure 8 shows TEM images of the precipitates interacting with dislocations in 0.7 wt.% Mn alloy after the tensile test at ambient temperature. The BF and DF (dark-field) images (see Figure 8a,b) reveal that the dislocations are clustered around and inside the T_Mn_ particles, resulting in the dislocation pinning effect, which can enhance strength. Hence, the T_Mn_ nanoprecipitate can impede the dislocation motion and absorb the strain energy as proficiently as the nanoscale precipitation [36]. A great number of the dislocations entangled and accumulated around the θ″ phase (Figure 8c). Considering that the θ″ phase is coherent with the α–Al matrix while the T_Mn_ phase is incoherent with the α–Al matrix, the θ″ phase follows the shearing mechanism, whereas the T_Mn_ phase follows the Orowan bypass mechanism [33,34]. The θ″ phase can increase the strength through interfacial strengthening, coherency strengthening, and modulus mismatch strengthening, which are derived from the interaction between the θ″ phase and the dislocation [37,38]. It is well known that a greater volume fraction of the θ″ and T_Mn_ precipitates results in a significant increase in strength; however, the T_Mn_ phase increases while the θ″ phase decreases as the Mn content increases in the studied alloys. Consequently, further calculation of the contribution to strength is necessary to elucidate the trend of strength variation.

### 4.3. Strengthening Mechanisms

With regard to strengthening mechanisms, the factors affecting the strength of Al–Cu–Mg–*x* Mn alloys should be associated with grain-boundary strengthening, solid solution strengthening, and precipitation hardening. The contributions of different strengthening mechanisms to the yield strength $\sigma_{YS}$ will be discussed in this section to explain the reasons for the strength changes related to the studied alloys. Here, 0 wt.%, 0.7 wt.%, and 1.1 wt.% Mn alloys are selected to elucidate strengthening mechanisms and the changes in experimental values of $\sigma_{YS}$. The $\sigma_{YS}$ can be estimated by [39,40]
(1)$$\sigma_{YS}=\sigma_{0}^{Al}+\sigma_{SS}+\sigma_{GB}+\sigma_{p}$$
where $\sigma_{0}^{Al}$ is the resistance to dislocation glide within the crystallite given by $\sigma_{0}^{Al}$~10 MPa [36], $\sigma_{SS}$ is the solid solution strengthening involved in solute atoms, $\sigma_{GB}$ is the grain-boundary strengthening, and $\sigma_{p}$ is the precipitation strengthening caused by precipitates. The solid solution strengthening can be quantitatively calculated as follows [41,42]:(2)$$\sigma_{SS}=H_{Cu}C_{Cu}+H_{Mg}C_{Mg}+H_{Mn}C_{Mn}$$
where $H_{Cu}$, $H_{Mg}$, and $H_{Mn}$ refer to the solution strengthening efficiencies and $C_{Cu}$, $C_{Mg}$, and $C_{Mn}$ refer to the mass fractions of the solute elements. The strengthening coefficients of the solute elements are $H_{Cu}$ = 13.8 MPa/wt.%, $H_{Mg}$ = 18.6 MPa/wt.%, and $H_{Mn}$ = 30.3 MPa/wt.% [42,43]. After solid solution treatment and artificial aging, some solute atoms are consumed to precipitate the T_Mn_ and θ″ phases. Therefore, EDS was used to determine the residual contents of Cu, Mg, and Mn in the matrix of the Al–Cu–Mg–*x* Mn alloys subjected to different Mn contents, as shown in Table 3. The contributions of solid solution strengthening to $\sigma_{YS}$ in 0 wt.%, 0.7 wt.%, and 1.1 wt.% Mn alloys are estimated based on the abovementioned formula, accounting for 65.8 MPa, 68.6 MPa, and 67.4 MPa, respectively.

The contribution of grain-boundary strengthening is usually calculated using the following Hall-Petch equation [43]:(3)$$\sigma_{GB}=k_{GB}d_{GB}^{-\frac{1}{2}}$$
where $k_{GB}$ is the Hall-Petch coefficient, ~0.15 MPa/$m^{1/2}$ [44], and $d_{GB}$ is the average grain size. Figure 9 shows the inverse pole figures and grain size statistical data of the Al–Cu–Mg–*x* Mn alloys. Based on the EBSD results, the average grain sizes in 0 wt.%, 0.7 wt.%, and 1.1 wt.% Mn alloys are 70.2 μm, 60.9 μm, and 52.4 μm, respectively. The results indicate that Mn has a refining effect on the grain size; however, it maintains the same order of magnitude. The contributions of grain-boundary strengthening to 0 wt.%, 0.7 wt.%, and 1.1 wt.% Mn alloys are estimated to be 17.9 MPa, 19.2 MPa, and 21.2 MPa, respectively.

The discrepancy in increasing mechanical strength caused by the T_Mn_ and θ″ phases of the experimental alloys can be explained using the equations for the strengthening mechanism. Multiple interactions are observed between the θ″ phases and dislocations, which are attributed to their coherence with matrix and shear ability, resulting in interfacial strengthening, modulus mismatch strengthening, order strengthening, and coherency strengthening. The reinforcing effects of order and modulus mismatch strengthening can be neglected based on prior research [34]. The increase in critically resolved shear stress (CRSS) caused by the interfacial strengthening of the θ″ phase is associated with the formation of new interfaces in the dislocation interactions, which can be calculated as follows [31,37]:(4)$$∆\tau =\left(\frac{0.908D_{p}}{t_{p}^{2}}\right)\cdot {\left(\frac{b\cdot f}{\Gamma }\right)}^{\frac{1}{2}}\cdot \gamma_{i}^{\frac{3}{2}}$$
where the interface energy is $\gamma_{i}$ = 0.21 J/m^2^, and the Burgers vector (b) for aluminum alloy is 0.286 nm. The dislocation line tension Γ can be represented as follows [33]:(5)$$\Gamma =\frac{Gb^{2}}{2\pi }\ln\sqrt{\frac{D_{p}^{2}}{2b^{2}f}}$$
where the shear modulus G of the α–Al matrix is equal to 28 GPa. The f, $D_{p}$, and $t_{p}$ values come from the aforementioned quantitative analysis results of precipitates, which are summarized in Table 2.

Given the coherency strengthening, the contribution of the θ″ phase to the CRSS can be estimated as follows [44]:(6)$$∆\tau =4.1\cdot G\cdot \left|\varepsilon^{\frac{3}{2}}\right|\cdot {\left[\frac{fD_{p}}{2b}\right]}^{\frac{1}{2}}$$
where the lattice strain ε = 0.006. The YS increment induced by precipitates in the Al–Cu–Mg–*x* Mn alloys can be obtained by [39]
(7)$$\sigma_{YS}=M\cdot ∆\tau$$
where the Taylor factor M = 3. The calculated YS increments are introduced into the above-mentioned equation, and the $\sigma_{YS}$ value of the Mn-free alloy is 186.2 MPa. The increments in $\sigma_{YS}$ resulting from the θ″ phases in the 0.7 wt.% and 1.1 wt.% Mn alloys are calculated using Equations (4)–(7), obtaining yield strengths of approximately 131.7 MPa and 97.2 MPa, respectively, which illustrates that the contribution of the θ″ phase decreases with the increase in Mn content.

In addition, the rod-like T_Mn_ phases that form during solution treatment in the Mn-containing alloys are non-shearable, which enhances the yield strength by the Orowan looping. The CRSS induced by the T_Mn_ phase can be expressed as follows [45,46]:(8)$$∆\tau =\frac{Gb}{2\pi \sqrt{1-v}}\cdot \frac{1}{\lambda }\cdot \ln\frac{\pi d_{p}}{{4r}_{0}}$$
where v is Poisson’s ratio equal to 0.33, $r_{0}$ is the core radius of the dislocation given by $r_{0}$ = b = 0.286 nm, and the mean planar inter-precipitate spacing λ is given as [47]
(9)$$\lambda =\left(C\sqrt{\frac{\pi }{6f_{v}}}-\frac{\pi }{4}\right)\cdot d_{p}$$
where the constant C takes a value of 1.23 for random arrays of precipitates. The increments in $\sigma_{YS}$, which are attributed to the T_Mn_ phase in 0.7 wt.% Mn and 1.1 wt.% Mn alloys, are estimated to be 69.4 MPa and 102.1 MPa, respectively.

Table 4 summarizes the contributions of different strengthening mechanisms to the $\sigma_{YS}$ of Al–Cu–Mg–*x* Mn alloys. The $\sigma_{YS}$ estimated by the strengthening model matches well with the experimental measurement values, accounting for over 84%. The contribution of solid solution strengthening and grain-boundary strengthening to the $\sigma_{YS}$ of the experimental alloys change slightly with increasing Mn content. Precipitation strengthening, from 186.2 MPa in the Mn-free alloy to 209.1 MPa in the 0.7 wt.% Mn alloy, primarily explains the reason for the improved $\sigma_{YS}$ of the Mn-containing alloys. However, the contribution of precipitation strengthening to $\sigma_{YS}$ arising from the θ″ and T_Mn_ nanoprecipitates decreases to 197.7 MPa compared to the 0.7 wt.% alloy. The results indicate that the increasement in the strength contribution of the T_Mn_ phase to $\sigma_{YS}$ partially compensates for the decrease in strength caused by the decreasing volume fraction of the θ″ phase.

## 5. Conclusions

In this work, the addition of various Mn contents on the microstructure evolution and mechanical properties of the as-cast and T6-state Al-Cu–Mg–*x* Mn alloys were investigated at ambient temperature. The microstructure of the Mn-containing alloys includes the primary eutectic T phase, a multitude of solid-solution-precipitated T_Mn_ particles, and a high density of nanoscale θ″ precipitates. However, the increasing Mn content promotes the formation of the T_Mn_ phase and inhibits the precipitation of the θ″ phase. Consequently, the synergistic precipitation of T_Mn_ and θ″ phase results in an equilibrium relationship between the volume fractions of the two phases, resulting in a combination of ultimate tensile strength of 497 MPa, yield strength of 340 MPa, and fracture elongation of 14% for 0.7 wt.% Mn alloy. Based on the contribution calculation of the strengthening mechanism, the impacts of grain-boundary strengthening, solid solution strengthening, and precipitation strengthening on the yield strength were quantitatively evaluated. Among them, precipitation of T_Mn_ and θ″ phase accounts for the prominent contribution to yield strength of Al–Cu–Mg–*x* Mn alloys.

## Figures and Tables

**Figure 1 nanomaterials-13-03038-f001:**
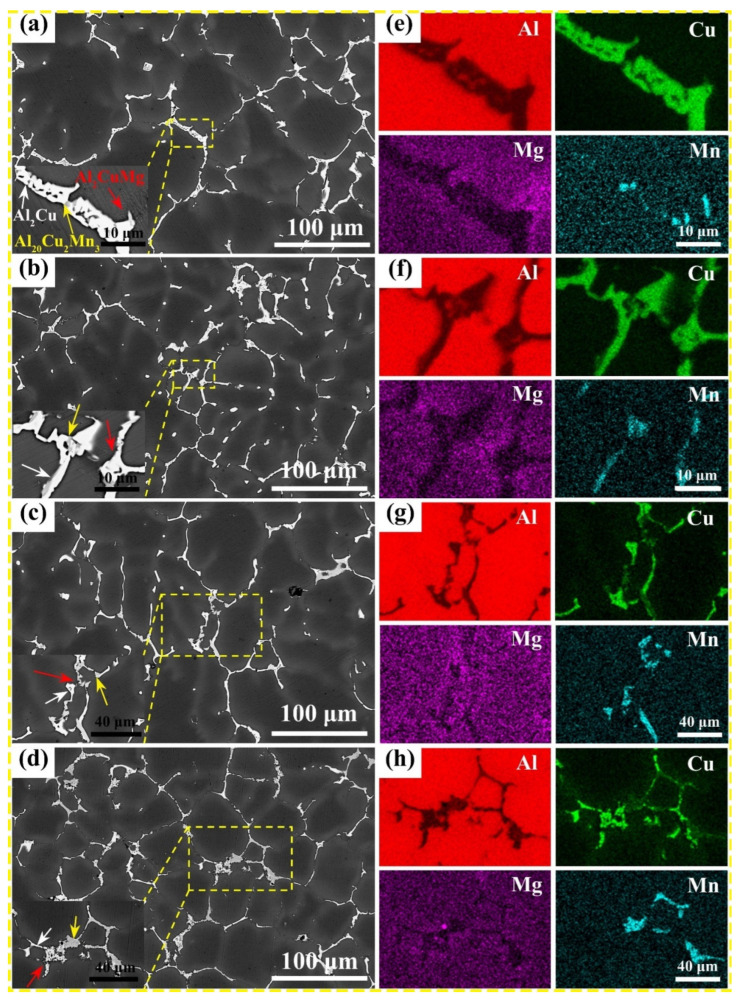
As-cast microstructure of Al–Cu–Mg–*x* Mn alloys with different Mn contents: (**a**,**e**) 0.5 wt.%, (**b**,**f**) 0.7 wt.%, (**c**,**g**) 0.9 wt.%, and (**d**,**h**) 1.1 wt.%; (**a**–**d**) backscattered electron images showing the eutectic morphology; and (**e**–**h**) elemental analysis results showing element distributions corresponding to the inserts in the lower left corner of (**a**–**d**).

**Figure 2 nanomaterials-13-03038-f002:**
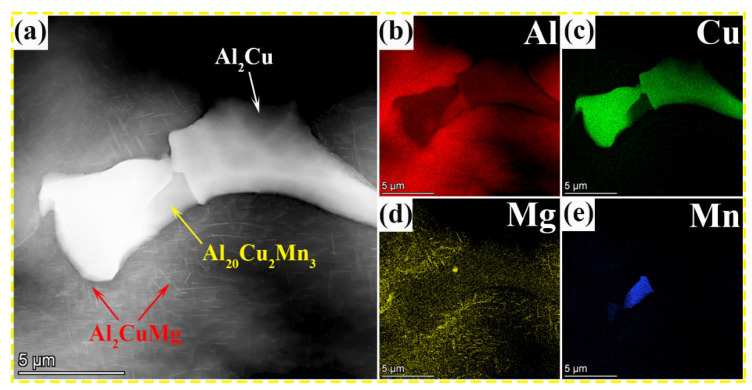
As-cast microstructure of 0.7 wt.% Mn alloy: (**a**) TEM HAADF (high-angle annular dark-field) image and (**b**–**e**) energy-dispersive spectroscopy (EDS) mappings showing constitution element distributions.

**Figure 3 nanomaterials-13-03038-f003:**
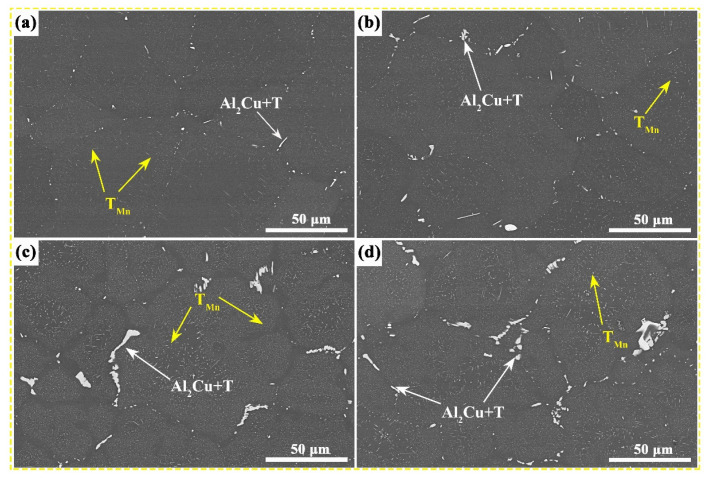
BSE images of the T6-state Al–Cu–Mg–*x* Mn alloys with different Mn contents: (**a**) 0.5 wt.%, (**b**) 0.7 wt.%, (**c**) 0.9 wt.%, and (**d**) 1.1 wt.%.

**Figure 4 nanomaterials-13-03038-f004:**
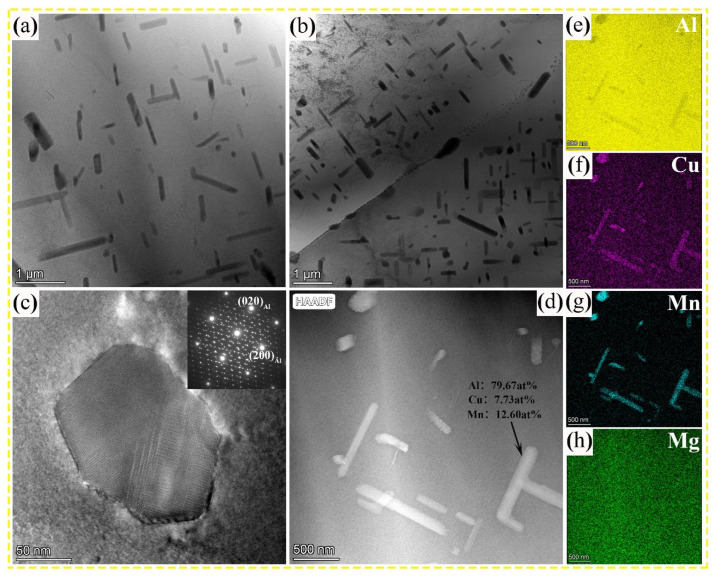
Investigation of the T_Mn_ precipitates: bright-field (BF) images containing the T_Mn_ phase of (**a**) 0.7 wt.% Mn alloy and (**b**) 1.1 wt.% Mn alloy; (**c**) the cross-section of the T_Mn_ phase and corresponding SAED pattern; (**d**) TEM HAADF image; and (**e**–**h**) EDS mappings showing constitution element distributions in (**d**).

**Figure 5 nanomaterials-13-03038-f005:**
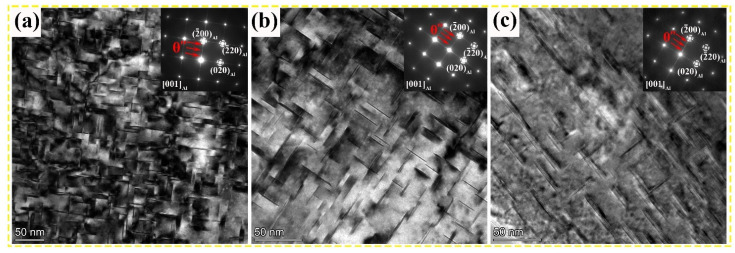
Representative BF images showing the θ″ precipitates in Al–Cu–Mg–*x* Mn alloys: (**a**) 0 wt.% Mn alloy, (**b**) 0.7 wt.% Mn alloy, and (**c**) 1.1 wt.% Mn alloy. The corresponding SAED patterns are given as inserts in the upper right corners of BF images in (**a**–**c**).

**Figure 6 nanomaterials-13-03038-f006:**
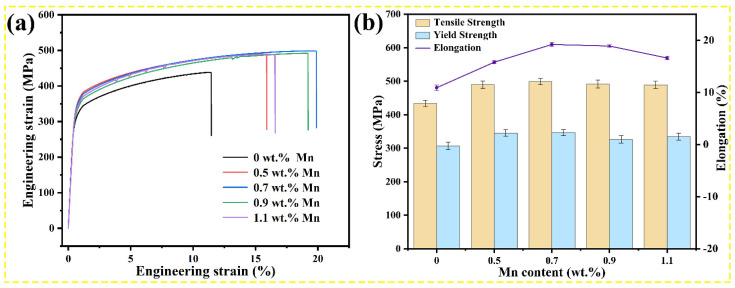
Mechanical properties of the Al–Cu–Mg–*x* Mn alloys: (**a**) tensile stress-strain curves at ambient temperature and (**b**) ultimate tensile strength, yield strength, and fracture elongation of alloys with different Mn contents.

**Figure 7 nanomaterials-13-03038-f007:**
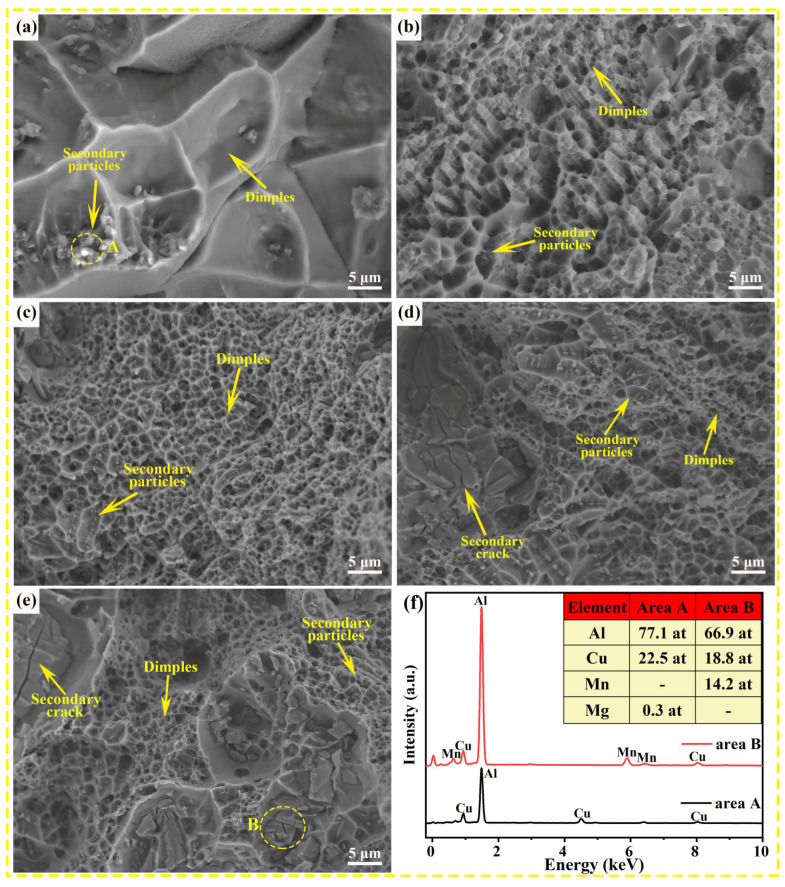
The fracture morphologies of Al–Cu–Mg–*x* Mn alloy with different Mn contents and composition analysis of secondary phases: (**a**) 0 wt.% Mn, (**b**) 0.5 wt.% Mn, (**c**) 0.7 wt.% Mn, (**d**) 0.9 wt.% Mn, and (**e**) 1.1 wt.% Mn; and (**f**) EDS results.

**Figure 8 nanomaterials-13-03038-f008:**
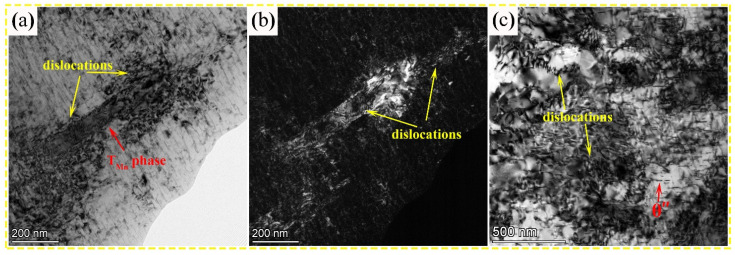
TEM images showing the interaction of the precipitates with dislocations in 0.7 wt.% Mn alloy after the tensile test at ambient temperature: (**a**,**b**) BF and DF (dark-field) images of the T_Mn_ phase and (**c**) BF image of the θ″ phase.

**Figure 9 nanomaterials-13-03038-f009:**
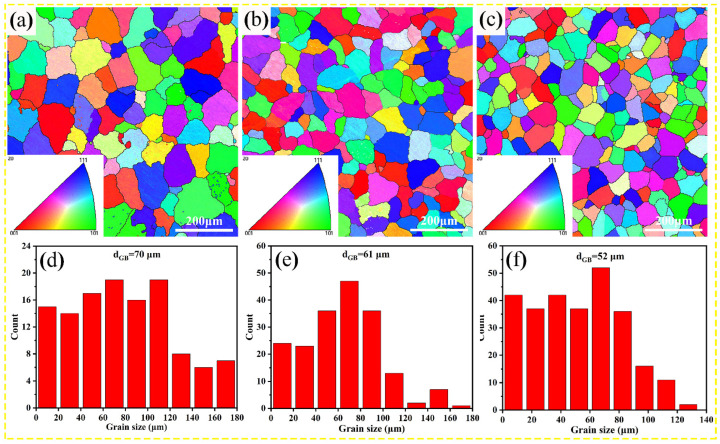
Inverse pole figure and grain size statistical data of the Al–Cu–Mg–*x* Mn alloys: (**a**,**d**) 0 wt.% Mn alloy, (**b**,**e**) 0.7 wt.% Mn alloy, and (**c**,**f**) 1.1 wt.% Mn alloy.

**Table 1 nanomaterials-13-03038-t001:** Chemical compositions of the Al–Cu–Mn–*x* Mg alloys (wt.%).

Alloys	Cu	Mn	Mg	Al
0 wt.% Mn	4.81	–	0.29	Bal.
0.5 wt.% Mn	4.89	0.41	0.27	Bal.
0.7 wt.% Mn	4.78	0.65	0.30	Bal.
0.9 wt.% Mn	4.86	0.83	0.33	Bal.
1.1 wt.% Mn	4.92	1.07	0.32	Bal.

**Table 2 nanomaterials-13-03038-t002:** Quantitative parameters of T_Mn_ and θ″ phases for the T6-state Al–Cu–Mg–*x* Mn alloys.

Alloy	Precipitates	$\mathrm{Mean}\;\mathrm{Length}\;D_{m}$ (nm)	$\mathrm{Mean}\;\mathrm{Thickness}\;t_{p}$ (nm)	$\mathrm{Number}\;\mathrm{Density}\;N_{v}$ (μm^−3^)	Volume Fraction f (%)
0 wt.% Mn	θ″	34.3	2.7	9165	1.88
0.7 wt.% Mn	T_Mn_	598.5	112.0	19.37	4.23
	θ″	28.3	2.7	6515	1.04
1.1 wt.% Mn	T_Mn_	576.4	124.7	27.3	6.01
	θ″	30.2	2.8	3352	0.65

**Table 3 nanomaterials-13-03038-t003:** Chemical compositions of the matrix in the Al–Cu–Mg–*x* Mn alloys.

Alloys	Elements (wt.%)			
	Al	Cu	Mg	Mn
0 wt.% Mn	95.35	4.31	0.34	–
0.7 wt.% Mn	95.28	4.03	0.33	0.36
1.1 wt.% Mn	95.46	3.77	0.36	0.41

**Table 4 nanomaterials-13-03038-t004:** Contributions of different strengthening mechanisms to the $\sigma_{YS}$ of the Al–Cu–Mg–x Mn alloys.

Alloys	$\mathrm{Solid}\;\mathrm{Solution}\;\mathrm{Strengthening},\;\sigma_{SS}$, MPa	Grain-Boundary $\mathrm{Strengthening},\;\sigma_{GB}$, MPa	$\mathrm{Precipitation}\;\mathrm{Strengthening},\;\sigma_{P}$, MPa	$\mathrm{Estimated}\;\sigma_{YS}$, MPa	$\mathrm{Experimental}\;\sigma_{YS}$, MPa
0 wt.% Mn	65.8	17.9	186.2	279.9	300.1
0.7 wt.% Mn	68.6	19.2	209.1	301.0	346.2
1.1 wt.% Mn	67.4	21.2	197.7	296.3	334.7

## Data Availability

All data in this study can be obtained by contacting the corresponding author.

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
