# Peer review of "Coupled Precipitation of Dual-Nanoprecipitates to Optimize Microstructural and Mechanical Properties of Cast Al–Cu–Mg–Mn Alloys via Modulating the Mn Contents"

_nanomaterials, 2023, doi:10.3390/nano13233038_

Round 1

Reviewer 1 Report

Comments and Suggestions for Authors

Nanomaterials-2705440

    In this paper, the effect of Mn content on the microstructure evolution and mechanical properties of Al– Cu–Mg–x Mn alloys at ambient temperature was investigated by changing the Mn contents. They obtained interesting results and will contribute to the industrial use of Al alloys.

   Aluminum alloy researchers are easy to understand the terms in this manuscript. But for readers of other fields, please add more explanations. For example, the crystal structures of T, S, θ, θ`, θ” and Ω phases. Also please explain T6 state. What is the difference between T phase and TMn phase? What is SSSS in Page 1.

So, I recommend this article for publication after minor revision.

1)      Title

“Co-precipitation” seems to be cobalt precipitation. I recommend that you do not use “Co-”.

2)  Page 3. 

What is the atmosphere during the melt-quenching process of Al alloys?

What is refining agent?

The solution treatment at 530C for 12 h seems a longer time as a standard T6 treatment. Are there any reasons?

3)  Page 4. “ The eutectic T phase is formed at the grain boundaries during solidification, whereas the TMn phase marked by yellow arrow is precipitated as a fine dispersion during the solution treatment.”

Why do you know that T phase is formed at the grain boundaries during solidification and TMn phase during solution treatment?

4)  Fig. 3 (a).

The grain-boundaries are not clear.

5)  Fig. 8 (a).

Is T TMn ?

Reviewer 2 Report

Comments and Suggestions for Authors

The authors have presented a comprehensive characterization of the influence of Mn content on the microstructure evolution and mechanical properties of Al–Cu–Mg–x Mn alloys. Their discussion is well-founded in both the obtained results and existing literature, demonstrating a strong scientific basis.

However, some minor revisions would further enhance the clarity and completeness of the manuscript before acceptance.

In Section 2.1, there's a discrepancy in temperature descriptions. The sequence of temperature changes, specifically cooling to 700°C and then reducing to 730°C, requires clarification for accuracy.

Regarding Section 2.2 on mechanical testing, it would be beneficial to detail the methodology used for strain measurement, particularly if an extensometer was employed. Additionally, specifying the gauge length and providing details on the load cell characteristics would strengthen this section.

For Section 2.3 on microstructure characterization, the description of specimen preparation could be improved. Notably, elaborating beyond grinding to include details about polishing to specific micrometer levels and specifying the etchants used would enhance clarity. Moreover, providing a more comprehensive description of the specimen preparation for TEM and including specifics on the EBSD analysis, such as scan size, step size, and rules for grain boundary identification, would be valuable additions.

Furthermore, in Figure 9, the caption and text mention EBSD inverse pole figure maps, not solely the inverse pole figure. It would be beneficial to include a color coding inverse pole figure triangle with the main orientations for clearer visualization.

As a suggestion for future work, exploring Kock-Mecking plots could offer a deeper understanding of strengthening behavior, aiding in comparative analysis and yielding identification.
